# Assessment of Ocean Bottom Pressure Variations in CMIP6 HighResMIP Simulations

Le Liu[1], Michael Schindelegger[1], Lara Börger[1], Judith Foth[1], and Junyang Gou[2]

[1]Institute of Geodesy and Geoinformation, University of Bonn, Bonn, Germany
[2]Institute of Geodesy and Photogrammetry, ETH Zurich, Zurich, Switzerland

**Correspondence:** Le Liu (le.liu@igg.uni-bonn.de)

**Abstract.** Ocean bottom pressure ($p_b$) variations from high-resolution climate model simulations under the CMIP6 (Coupled Model Intercomparison Project Phase 6) HighResMIP protocol are potentially useful for oceanographic and space-geodetic research, but the overall signal content and accuracy of these $p_b$ estimates have hitherto not been assessed. Here we compute
monthly $p_b$ fields from five CMIP6 HighResMIP models at 1/4° grid spacing over both historical and future time spans and compare these data, in terms of temporal variance, against observation-based $p_b$ estimates from a 1/4° downscaled GRACE (Gravity Recovery and Climate Experiment) product and 23 bottom pressure recorders, mostly in the Pacific. The model results are qualitatively and quantitatively similar to the GRACE-based $p_b$ variances, featuring—aside from eddy imprints—elevated amplitudes on continental shelves and in major abyssal plains of the Southern Ocean. Modeled $p_b$ variance in these regions is
~10–80% higher and thus overestimated compared to GRACE, whereas underestimation relative to GRACE and the bottom pressure recorders prevails in more quiescent deep-ocean regions. We also form variance ratios of detrended $p_b$ signals over 2030–2049 under a high-emission scenario relative to 1980–1999 for three selected models and find statistically significant increases of future $p_b$ variance by ~30–50% across deep Arctic basins and the southern South Atlantic. The strengthening appears to be linked to projected changes in high-latitude surface winds and, in the case of the South Atlantic, intensified
eddy kinetic energy. The study thus points to possibly new pathways for relating observed $p_b$ variability from (future) satellite gravimetry missions to anthropogenic climate change.

## 1 Introduction

Ocean bottom pressure ($p_b$) measures the weight of the total water column exerted on the seafloor, and thus its variations are inherently linked to mass fluxes and redistribution in the Earth system. For example, contemporary increases in ocean mass
and thus static $p_b$ result from loss of land ice and continued groundwater depletion (Ludwigsen et al., 2024). Dynamic $p_b$ signals, on the other hand, are largely driven by atmospheric forcing (e.g., Ponte, 1999; Boening et al., 2011; Petrick et al., 2014; Fukumori et al., 2015), but may also arise internally within the ocean through non-linear processes (Zhao et al., 2021). To the extent the ocean is in geostrophic balance, horizontal $p_b$ gradients can be used to infer barotropic transport variability of deep currents (e.g., Makowski et al., 2015) or characteristics of the meridional overturning circulation (Hughes et al., 2018).
Physical oceanography aside, knowledge of $p_b$ variability is also essential in geodesy, particularly for de-aliasing purposes in

satellite gravimetry processing (Shihora et al., 2022; Schindelegger et al., 2021). In addition, ocean mass redistributions excite Earth rotation changes (e.g., Harker et al., 2021) and cause crustal loading signals that may mask geophysically more relevant effects in station position time series (van Dam et al., 2012).

Works in the noted disciplines have been served well by $p_b$ estimates on a global scale from time-resolved satellite gravimetry (i.e., the Gravity Recovery and Climate Experiment, GRACE) and numerical ocean models. Pertinent model diagnostics are typically drawn from data-constrained or free simulations of the real ocean (e.g., Piecuch et al., 2015; Hughes et al., 2018; Androsov et al., 2020; Shihora et al., 2022; Ponte et al., 2024). With a few exceptions (Ponte et al., 2002; Bingham and Haines, 2005; Landerer et al., 2007), coupled climate models have so far played no role in studies of $p_b$ variability and its implications. However, such synthetic mass change records are potentially useful as background fields in future gravimetry mission simulations to assess the error characteristics and science return of specific satellite constellations (Daras et al., 2024). Moreover, climate models can be used to delineate likely changes in surface mass variability that may form target signals for future satellite gravity missions (cf. Jensen et al., 2020). Whether or not there are any significant changes in dynamic $p_b$ to expect in a warming world is an interesting question in its own right.

We are thus motivated to take a first look at $p_b$ estimates from the latest realization of the Coupled Model Intercomparison Project (CMIP), i.e., CMIP6. Specifically, our objectives are to (i) validate the $p_b$ diagnostics from CMIP6 experiments by comparing them in, a statistical sense, to observations, (ii) identify source models that could possibly serve as input for end-to-end satellite simulations, and (iii) test for likely future changes in $p_b$ variability under a high-emission scenario. Although the coarse-resolution (~100-km grid spacing) branch of CMIP6 would suffice most of these purposes, we here analyze the output of eddy-permitting (~25-km) ocean models under the CMIP6 HighResMIP (CMIP6-HR) protocol. Horizontal model resolution in this range is generally advantageous for representing topographically constrained dynamics (e.g., Harker et al., 2021). It also admits transient eddy features, which have their own unique imprint on $p_b$ (Bingham and Hughes, 2008; Hughes et al., 2018; Zhao et al., 2021). Eddy activity, in turn, is projected to change profoundly in the 21st century (Beech et al., 2022) and therefore may strengthen or weaken $p_b$ variability in certain locations. In total, and for practical reasons, we consider five out of the seven complete institutional contributions to the CMIP6-HR activity (cf. Haarsma et al., 2016; Roberts et al., 2020). The specific time periods examined are 1980–2014, representing recent historical climatic conditions, and 2030–2049 in the near future. Given the monthly sampling of the data and the possibility of spurious model drifts (Sen Gupta et al., 2013), we are precluded from studying trends and exclusively analyze $p_b$ signals at intraseasonal to interannual periods.

## 2 Data and methods

### 2.1 Climate model data

CMIP6-HR is an effort dedicated to the investigation of the role of horizontal resolution in CMIP-type climate simulations (Haarsma et al., 2016), see, e.g., Docquier et al. (2019), Roberts et al. (2020) or Shi et al. (2021) for evaluations of these simulations with respect to the ocean. We use the model outputs from the coupled historical runs (labeled as 'hist-1950' in HighResMIP), complemented by coupled scenario simulations ('highres-future') in several cases. The hist-1950 simulations

**Table 1.** Selected specifications of the CMIP6-HR models used in this study[a]

| | ECMWF-IFS | HadGEM3-GC31 | CNRM-CM6-1 | CMCC-CM2 | EC-Earth3P |
|---|---|---|---|---|---|
| Short name | ECMWF | HadGEM | CNRM | CMCC | Earth3P |
| Resolution label | HR | HM | HR | VHR4 | HR |
| Atmospheric model[b] | IFS cycle 43r1 | UM GA7.0 | ARPEGE-Climat6.3 | CAM4 | IFS cycle 36r4 |
| Atmosphere resolution in km | 25 | 50 | 35 | 25 | 50 |
| *Ocean/sea ice* | | | | | |
|   Ocean model | NEMO3.4 | NEMO3.6 | NEMO3.6 | NEMO3.6 | NEMO3.6 |
|   Horizontal resolution in ° (km) | 1/4° (25 km) | 1/4° (25 km) | 1/4° (25 km) | 1/4° (25 km) | 1/4° (25 km) |
|   No. vertical levels | 75 | 75 | 75 | 50 | 75 |
|   Variant label | r1i1p1f1 | r1i1p1f1 | r1i1p1f2 | r1i1p1f1 | r3i1p2f1 |
| Reference | Roberts et al. (2018) | Roberts et al. (2019) | Voldoire et al. (2019) | Cherchi et al. (2019) | Haarsma et al. (2020) |

[a] The analysis window is 1980–2014 (hist-1950 experiment) and, in addition, 2030–2049 for the HadGEM, CNRM, and Earth3P highres-future experiments under SSP5-8.5.

[b] Abbreviations: IFS (Integrated Forecasting System of the ECMWF), UM GA7.0 (Global Atmosphere 7.0 configuration of the Met Office Unified Model), ARPEGE (Action de Recherche Petite Echelle Grande Echelle), CAM4 (Community Atmosphere Model Version 4).

cover the period 1950–2014 and provide the initial conditions to highres-future, nominally integrated to the year 2050. External forcing fields comprise time-varying solar, volcanic, anthropogenic aerosol, ozone depletion, and greenhouse gas effects (Haarsma et al., 2016). For the highres-future simulations, these forcings follow the Representative Concentration Pathway 8.5 (i.e., high-end emission) scenario with Shared Socioeconomic Pathway 5 (SSP5, O'Neill et al., 2016) conditions, abbreviated as SSP5-8.5.

The five CMIP6-HR models examined in our work are ECMWF-IFS-HR (European Centre for Medium-Range Weather Forecasts Integrated Forecasting System), HadGEM3-GC3.1 (Hadley Centre Global Environment Model 3 – Global Coupled vn 3.1), CNRM-CM6-1 (Centre National de Recherches Météorologiques Climate Model version 6), CMCC-CM2 (Euro-Mediterranean Centre on Climate Change Coupled Climate Model version 2), and EC-Earth3P; see Table 1 for an overview. Abbreviations, used throughout the paper, are ECMWF, HadGEM, CNRM, CMCC, and Earth3P, respectively. Given the scope of our study, we focus on the high-resolution (label HR, HM, or VHR4) configuration of each model, comprising horizontal grid spacings of 1/4° (∼25 km) for the ocean and 25 to 50 km for the atmosphere. The selected models differ with regard to the atmospheric component but not for the ocean (Table 1); the ocean component in all five cases is NEMO (Nucleus for European Modelling of the Ocean, either version 3.4 or 3.6, Madec, 2016), a hydrostatic, primitive equation general circulation model. Processes and properties of the ocean are therefore represented very similarly in all five climate models. Hence, any

spread in $p_b$ variability reported below rather bears on parameter uncertainty, internal variability, or structural uncertainty in the atmospheric component. Values for selected parameters of the ocean component (e.g., viscosity and diffusivity) can be found in Docquier et al. (2019) and Roberts et al. (2020).

To deduce each model's bottom pressure, we retrieve monthly mean fields for potential temperature (standard name 'thetao'), salinity ('so'), and sea surface height above geoid ('zos'). Alternatively, we could use the models' diagnostic $p_b$ output (standard name 'pbo') without invoking assumptions in the calculation of $p_b$; cf. Sect. 2.2. Not all of the tested models provide 'pbo' output, though. In any event, differences between the original and our inferred $p_b$ values in terms of their temporal RMS (root-mean-square) are very small (Supplementary Fig. S1). For the hist-1950 experiments, we focus on the last 35 simulation years (1980–2014). The time span considered for the highres-future simulations is 2030–2049. These choices were made in view of manageable data volumes, data availability, and the requirement for extended overlap with the $p_b$ time series from satellite gravimetry (Sect. 2.3). For complementary analyses and tentative interpretation of the $p_b$ results, we additionally process monthly eastward and northward seawater velocities ($u$, $v$; standard names 'uo', 'vo') and the eastward and northward components of the near-surface atmospheric wind ('uas', 'vas', usually referred to an altitude of 10 m).

## 2.2 Calculation of ocean bottom pressure anomalies

A mathematical expression for $p_b$, obtained by integrating the hydrostatic balance over full depth along the vertical coordinate $z$ (positive upward), is (e.g., Gill and Niller, 1973; Ponte, 1999)

$$p_b = \int_{-H}^{0} \rho g \mathrm{d}z + \int_{0}^{\eta} \rho g \mathrm{d}z \approx \int_{-H}^{0} \rho g \mathrm{d}z + \rho_0 g \eta \tag{1}$$

where $\rho = \rho(z)$ is the (in situ) density of seawater with surface value $\rho_0$, $\eta$ represents the sea level anomaly, $H$ is the local water depth, and $g$ is the acceleration due to gravity. While atmospheric pressure contributes to $p_b$ in nature, it is not part of the exchange variables between atmospheric and oceanic components of CMIP6-HR and therefore neglected in Eq. (1). We perform the vertical integration on the native model grid and specifically derive the value of $\rho$ at a given location from the modeled potential temperature and salinity, using the Gibbs SeaWater Oceanographic Toolbox (McDougall and Barker, 2011). Exact evaluation of Eq. (1) from CMIP6-HR data would also require each model's internal representation of bottom topography, e.g., in the form of a three-dimensional array of wet and dry cell fractions (or partial cell thicknesses; see Adcroft et al., 1997; Bernard et al., 2006). Given that these fractions are not provided in the CMIP6 archive, we calculate approximate versions of them based on given layer thicknesses and the ETOPO1 bathymetry dataset (Amante and Eakins, 2009), interpolated to the horizontal model grid. Although the information on the provenance of model bathymetries in CMIP6-HR is uncertain, the HadGEM ocean component description (Storkey et al., 2018) explicitly points to ETOPO1, and that is indeed a standard choice for NEMO simulations on the ORCA025 grid (cf. Börger et al., 2023).

We reduce the full bottom pressure to dynamic anomalies (also denoted $p_b$) by subtracting the time-mean pressure and trend values at each grid point, along with the monthly-varying spatial average of $p_b$. Global ocean mass changes, which may have physical and non-physical causes in numerical ocean models (e.g., Sen Gupta et al., 2013), are therefore not part of the

105 analyzed $p_b$ variability. To convey some sense of how this variability is distributed across frequencies while also keeping the description of the results compact, we consider (i) full-time series of detrended dynamic $p_b$ and (ii) non-seasonal anomalies obtained by removing least-squares fits of the annual and semiannual oscillations from the full-time series. Variant (ii) thus emphasizes intraseasonal and interannual $p_b$ signals. The seasonal cycle, which has been the subject of several recent GRACE and ocean model studies (e.g., Cheng et al., 2021; Chen et al., 2023; Ponte et al., 2024), is only treated in passing. Note that $p_b$

has standard SI units (Pa) in our calculations but is expressed as centimeters of equivalent water height (EWH) below, assuming the same references density (1025 kg m$^{-3}$) as in the GRACE data.

All comparisons to observations are conducted based on measures of signal variance, i.e., the temporal RMS or quantities derived thereof. As will be shown below (Sect. 3.5), the $p_b$ RMS at a given location is generally a non-stationary statistic that fluctuates under both constant and evolving climatic conditions. These fluctuations may render modeled and observed

$p_b$ variability statistically indistinguishable from each other. We indicate such cases in our presentation by testing whether the GRACE-based $p_b$ RMS is inside the 10th–90th percentile range of an approximate distribution of model RMS values at the considered grid point. For each model, the distribution is constructed by computing RMS values in rolling 15-year windows, progressively shifted forward by one year from 1980–1994 to 1990–2014. We thus have 21 RMS estimates, drawn from windows that are similar in length (although somewhat shorter) than the period of the adopted GRACE data. In figures

pertaining to the historical simulations, we always plot the median RMS of each model's distribution, which closely agrees with the RMS estimate over the full 1980–2014 period (typically within 0.2 cm).

## 2.3 GRACE data

The primary dataset for validating the CMIP6-HR bottom pressures is a high-resolution satellite-based $p_b$ product (Gou et al., 2025), obtained by downscaling available monthly $p_b$ anomalies from GRACE/-Follow On (Tapley et al., 2019) beyond their

typical effective resolution of $\sim$3°. This product, called GRACE-DS hereinafter, offers $p_b$ information from April 2002 to December 2020 on a $1/4° \times 1/4°$ grid. The downscaling was realized using a self-supervised deep-learning algorithm, guided by $p_b$ diagnostics from two eddy-permitting ocean reanalyses based on the NEMO model (Lellouche et al., 2013; Zuo et al., 2017). Spatial detail on sub-GRACE grid scales was obtained by maximizing similarities between the downscaled product and the reanalysis output, while constraining the solution to the original GRACE values at larger scales (Gou et al., 2025). The

data fusion algorithm was further enhanced by supervision signals based on input features, comprising, e.g., wind stress and bathymetry. From the three downscaled solutions provided by Gou et al. (2024), we use the one based on GRACE mascons from the Center for Space Research (CSR, RL06.2, Save et al., 2016).

A visual comparison of the GRACE-DS and CSR fields in terms of the RMS of dynamic, detrended $p_b$ anomalies is provided in Fig. 1. As per design, the deep-learning algorithm retains the signal content of the CSR parent solution at wavelengths of a

135 few hundred kilometers and longer. At the same time, short-scale features are added throughout the ocean, including marked $p_b$ increases in eddy-active regions (e.g., Southern Ocean, western boundary current regions). The downscaling approach also enhances near-shore amplitudes on several continental shelves, usually by 20–30% (e.g., around Australia, China, the Atlantic coast of South America, or on the Siberian Shelf). As these spatial details are mainly derived from the NEMO-based ocean

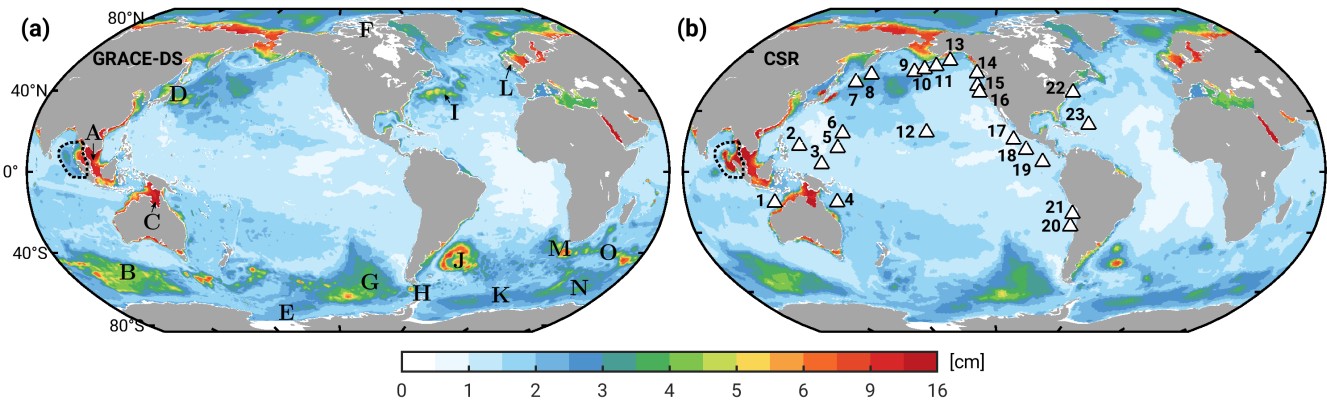

**Figure 1.** RMS of monthly $p_b$ anomalies (in cm of EWH) from (a) GRACE-DS, obtained by downscaling $1°$ CSR mascons (Save et al., 2016) to a $1/4° \times 1/4°$ grid. The RMS of the CSR solution is illustrated in panel (b). Analyzed time series are from April 2002 to December 2020, with trends and spatial mean $p_b$ signals removed. White triangles in (b) indicate the 23 BPR stations used in this study, labeled by increasing longitude. The dashed black polygon in the Andaman Sea marks the region excluded from the comparison in Sect. 3 due to the presence of seismic deformation signals in the GRACE fields. Lettered keys in panel (a) denote the (**A**) Gulf of Thailand, (**B**) Australian Antarctic Basin, (**C**) Gulf of Carpentaria, (**D**) Kuroshio Current, (**E**) Ross Sea, (**F**) Canadian Arctic Archipelago, (**G**) Bellingshausen Basin, (**H**) Drake Passage, (**I**) Gulf Stream Extension, (**J**) Argentine Basin/Zapiola Gyre, (**K**) Weddell Basin, (**L**) Celtic Sea, (**M**) Cape Basin/Agulhas leakage, (**N**) Enderby Basin, (**O**) Agulhas Return Current.

reanalyses, the model-data comparison in eddying regions and near continental boundaries should to be treated with caution. In the supplementary to this article, we thus repeat parts of our model assessment using (i) the CSR RMS map in Fig. 1b, and (ii) another $p_b$ mascon solution from Jet Propulsion Laboratory (JPL, RL06.3M, Watkins et al., 2015; Wiese et al., 2023). Whereas CSR employs the same $1/4° \times 1/4°$ grid as GRACE-DS, the JPL mascons are discretized in the form of $3° \times 3°$ equal-area caps. Prior to comparing to JPL data, we average the model $p_b$ fields to the exact geometry of these caps (cf. Ponte et al., 2024). Such coarsening suppresses most of the mesoscale imprints on $p_b$, which are, in any case, difficult to assess. For the presentation in the main text, we focus on GRACE-DS, as its characteristics are most commensurate with the CMIP6-HR models across scales and from deep to shallow regions.

Sources of inconsistency between CMIP6-HR and the GRACE(-DS) data, among other effects, the absence of barometric pressure-driven $p_b$ variability in the models (cf. Piecuch et al., 2022, $\lesssim 0.5$ cm in places) and seismic gravity/deformation signals associated with major earthquakes in the gravity fields solutions (e.g., Zhang et al., 2020). The largest such spurious contribution to $p_b$, apparent in Fig. 1 around the Malay Peninsula, stems from the 2004 Sumatra-Andaman and 2012 Indian Ocean earthquakes. We approximately delimit the affected region based on insight from literature (De Linage et al., 2009) and enhanced ($\geq 2$ cm) open-ocean $p_b$ values, and exclude it from all direct comparisons to the model fields.

## 2.4 In situ observations

We also compare the CMIP6-HR $p_b$ diagnostics with direct seafloor pressure measurements from the DART® (Deep-ocean Assessment and Reporting of Tsunamis) program (Bernard and Meinig, 2011). DART is a US-led tsunami early warning system comprising more than 50 active deep-ocean bottom pressure recorders (BPRs) located mainly along the Circum-Pacific belt. The DART data are disseminated in meters of EWH with a nominal sampling of 15 min. From the full network, we extract 23 stations with nearly continuous observations over an 11-year time span from 2013 to 2023. The year 2013 is a reasonable start date, as it marks the completion of a major upgrade to the DART technology (Bernard and Meinig, 2011). Each of the 23 time series is a concatenation of recordings from successive deployments in slightly different locations, typically differing by O(10 m) in the overlying pressure. We process each of these segments individually, removing (i) 3 days of observations at the start and end of the time series, (ii) a fitted second-order polynomial (to account for sensor drift, cf. Poropat et al., 2018), (iii) occasional spikes, and (iv) signals of 16 largest tidal constituents (Hart-Davis et al., 2021, including fortnightly, monthly, and semiannual constituents). Steps (i)–(iv) are applied to hourly-sampled versions of the segments, which we subsequently splice together and average into a monthly $p_b$ series per station.

For consistency with our treatment of the CMIP6-HR fields (Sect. 2.1), the DART data should also be cleaned from global mean $p_b$ signals, reflecting both net atmospheric pressure variations over the ocean and true ocean mass changes (Ponte, 1993; Johnson and Chambers, 2013). While the former is readily corrected for using monthly sea level pressures from a modern atmospheric reanalysis (Hersbach et al., 2020), precise estimates for the mean ocean mass change over 2013–2023 are harder to come by. However, apart from a trend, these changes mainly consist of an annual oscillation, reaching ∼0.5 cm in amplitude (Johnson and Chambers, 2013). We thus remove from the detrended DART data a synthetic time series of the seasonal cycle in ocean mass, constructed from $p_b$ output (1992–2017) of an ocean state estimate (the Estimating the Circulation and Climate of the Ocean, Version 4 Release 4b, ECCOv4rb, Forget et al., 2015). ECCOv4rb is deemed suitable for this purpose, as it is constrained to most available oceanographic data, including GRACE-based $p_b$ and observed global ocean mass changes.

Our final network, illustrated in Fig. 1b, comprises 23 BPRs featuring a mean cumulative record length of 10.8 years, with monthly averages typically built from 260 to 350 days of valid observations per year. The in situ time is, therefore, much shorter than the CMIP6-HR records but sufficiently long to constrain variability at the annual and higher frequencies (cf. Ray et al., 2021, for the case of sea level). Interannual $p_b$ variability, which is less well represented by these ∼10-year records, accounts for about 30% of the variance in the full monthly DART time series. The network also heavily emphasizes the Pacific, with only two stations in the western North Atlantic and one sensor located in the Indian Ocean (North Australian Basin). We nevertheless prefer to perform our analysis based on a homogeneous, consistently processed in situ dataset and refrain from adding other BPR time series, which are usually short in duration (≲ 1–4 years, Poropat et al., 2018; Androsov et al., 2020; Schindelegger et al., 2021) or confined to special locations (e.g., the Arctic Beaufort Gyre, Kemp et al., 2005).

## 3 Results

### 3.1 Overview of modeled $p_b$ variability

Figure 2 (left column) illustrates the RMS of monthly dynamic $p_b$ anomalies, calculated for each of the five CMIP6-HR models over the period 1980–2014 (as median of 21 RMS values in rolling 15-year windows). The model results are qualitatively and quantitatively very similar to the $p_b$ variability in the GRACE-DS product (Fig. 1a), ocean state reconstructions (Piecuch et al., 2015; Gou et al., 2025), and ocean forward models forced by atmospheric analysis fields (e.g., Hughes et al., 2018; Poropat et al., 2018; Androsov et al., 2020). Among the defining features are a general increase in $p_b$ variability from temperate to polar latitudes, elevated amplitudes in several marginal seas (e.g., Hudson Bay, Mediterranean Sea, Red Sea), and largest signals near continental boundaries. Shallow depths facilitate a vigorous barotropic response to the imposed wind stress (Vinogradova et al., 2007), leading to peak values of 10–15 cm in Fig. 2 over, e.g., the Siberian Shelf, Chukchi Sea, Gulf of Thailand, Gulf of Carpentaria, and Baltic Sea. In some of these places, coastally trapped waves contribute to the spreading of locally generated mass anomalies (e.g., Oliver and Thompson, 2011; Fukumori et al., 2015).

In the deep ocean, known regions of relatively high barotropic variability are the Arctic Ocean, the subpolar Northwest Pacific (Petrick et al., 2014), and the three major abyssal plains in the Southern Ocean (Bellingshausen, Australian-Antarctic, and Weddell-Enderby basins). These plains are encased by nearly closed contours of planetary potential vorticity, thus trapping and amplifying forced barotropic circulations (e.g., Fukumori et al., 1998; Weijer and Gille, 2005; Ponte and Piecuch, 2014; Weijer, 2015). The Australian-Antarctic basin is the most energetic of the three regions, with RMS values of ∼4–6 cm in all five models. Bottom topography also plays a key role in generating the characteristic, near-uniform fluctuation in $p_b$ (up to 4 cm, Fig. 2) in the interconnected deep basins of the Arctic Ocean and Nordic Seas. In detail, wind stress along the continental slope results in cross-slope Ekman transport, which creates $p_b$ anomalies of opposite sign between the shallow shelf and the adjacent deep ocean (Fukumori et al., 2015). The deep-ocean mass anomaly propagates away from its source regions as trapped Kelvin wave, separates from its shallow counterpart at straits and sills, and eventually equilibrates across the deep Arctic basins that are bounded by gradients of potential vorticity.

Forced barotropic dynamics aside, the CMIP6-HR models also contain an active mesoscale field, which can modulate the $p_b$ diagnostics on local to basin-wide scales at intraseasonal and longer periods (Hughes et al., 2018; Zhao et al., 2021, 2023). Localized imprints of eddies and instabilities are most obvious in western boundary current regions (Agulhas Retroreflection, Kuroshio, Gulf Stream) and broadly along the path of the Antarctic Circumpolar Current (Androsov et al., 2020). A stand-out feature in all models, perhaps with the exception of CMCC (Fig. 2d), is a ∼8-cm G-shaped $p_b$ structure in the Argentine Basin. The pattern is the signature of the Zapiola Anticyclone, a well-known recirculation supported by high eddy activity in the western parts of the basin and interactions of that eddy field with bottom topography (de Miranda et al., 1999; Hughes et al., 2007). The GRACE-DS product suggests a similarly G-shaped structure around the Zapiola Drift (Fig. 1a), albeit inherited from the utilized ocean reanalyses and not the GRACE gravity fields themselves (Fig. 1b).

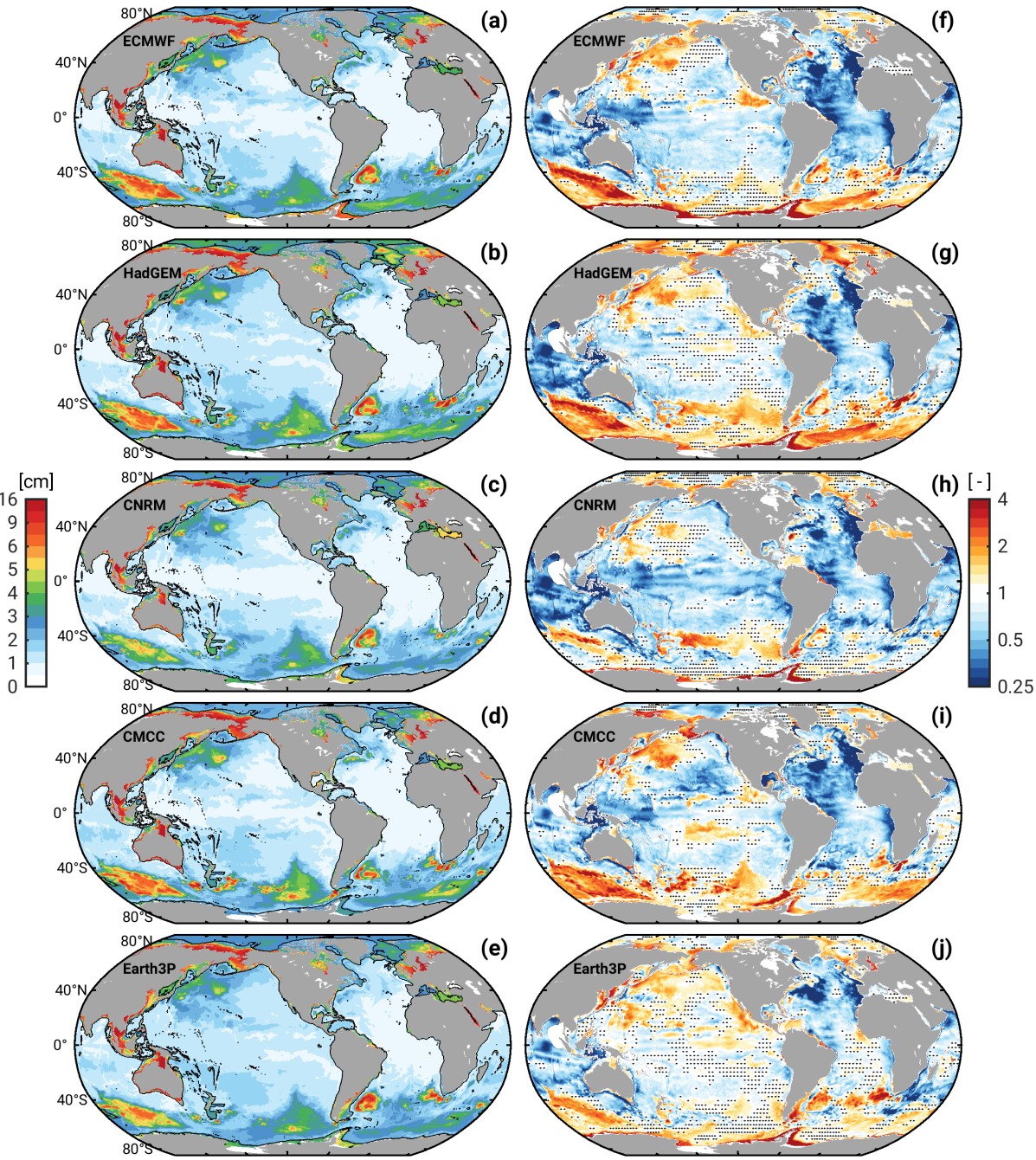

**Figure 2.** Plots on the left show the RMS (cm) of detrended $p_b$ time series from five CMIP6-HR models, computed as median of the RMS distribution over 1980–2014, see Sect. 2.2 for details. Plots on the right show each model's variance ratio, $R$, relative to the $p_b$ variance from GRACE-DS (i.e., $R > 1$ indicates higher bottom pressure variability in the model). Stippling marks $3° \times 3°$ cells where the RMS of GRACE-DS (Fig. 1a) is within the 10th–90th percentile range of the model RMS for at least a third of the contained $1/4°$ grid points. The black solid line is the 1000-m isobath, and the region of the Andaman Sea anomaly seen in GRACE-DS is masked out.

## 3.2 Comparison with GRACE – broadband variability

For a more quantitative analysis of the agreement between modeled and observation-based $p_b$ fluctuations, we compute variance ratios, $R$, defined as

$$R = \frac{\sigma^2_{mod}}{\sigma^2_{obs}} \qquad (2)$$

where $\sigma^2_{mod}$ represents the $p_b$ variance of a chosen model and $\sigma^2_{obs}$ is the observed variance at a given location, calculated from
220 GRACE-DS, CSR or JPL mascons, or the BPRs.

Figure 2 (right column) illustrates estimates of $R$ for the full (i.e., unfiltered) $p_b$ model time series relative to GRACE-DS, deduced after interpolating $\sigma^2_{mod}$ from the native model grid to the GRACE-DS grid. Despite differences in detail, the five maps have several features in common. In particular, they suggest a tendency for excess model variance ($R > 1$) in areas of enhanced $p_b$ amplitudes (Fig. 2, left column), while underestimation by the models ($R < 1$) prevails in less energetic regions.
One may argue that the latter is simply due to noise in the chosen GRACE product, dominating the low and mid-latitude RMS in Fig. 1a. However, random errors in a single monthly gravity field solution are typically estimated to be 3–4 cm (e.g., Kvas et al., 2019), which limits the noise component in the temporal RMS in Fig. 1 to 0.3 cm (inferred from a pessimistic variance propagation over 192 monthly $p_b$ fields). The prevalence of low variance ratios outside energetic regions is therefore more likely the result of systematic effects in the GRACE-DS product (e.g., imperfect corrections for low-degree zonal spherical
harmonics, Gałdyn and Sośnica, 2024) and structural errors in the CMIP6-HR models. Thoughts on possible model limitations are offered in Sect. 3.4. Out of the five cases considered, HadGEM and Earth3P compare most favorably to GRACE-DS; global median variance ratios, $\overline{R}$, deduced from all grid points with a model RMS of $\leq 3$ cm and in depths greater than 1000 m, are 0.81 (HadGEM) and 0.90 (Earth3P). These numbers are somewhat closer to 1 than for the other three models ($\overline{R} \approx 0.72$, Table 2).
Shifting the focus to the red (i.e., $R > 1$) patches in Fig. 2, we find that all models have higher $p_b$ variance than GRACE-DS in the Northwest Pacific, the Australian-Antarctic Basin, and (to some extent) the Weddell-Enderby abyssal plain. Excess values are least pronounced in CNRM and largest in CMCC, evident also from the global median variance ratios taken over deep energetic regions ($\overline{R} = 1.15$ vs. $\overline{R} = 1.75$, see Table 2). Excessive $p_b$ variance in the CMIP6-HR models is also seen for most continental shelf regions—in sub-polar and polar latitudes more so than in the tropics (cf., e.g., the Gulf of Thailand
and Indonesian Seas). This geographical dependence is further brought out in Table 2, where we list global median variance ratios in shallow water ($H < 1000$ m) both for latitudes higher and lower than $60°$ (a somewhat arbitrarily chosen cut-off). The overestimation of shallow $p_b$ variations within $60°$ from the equator is rather subtle, bounded by $\overline{R} = 1.30$ for HadGEM. The HadGEM value increases to $\overline{R} = 1.95$ in polar latitudes, followed by $\overline{R} \approx 1.40$ in the other models. Common regions of overly strong $p_b$ variations are the East Siberian Shelf, the Bering Sea, and the Antarctic continental shelf.
As for the $p_b$ imprints of mesoscale eddies, areas of high and low variance ratios tend to change from model to model, see, e.g., the Drake Passage or the Agulhas Retroreflection and its leakage to the west. However, most eddy-related variability in the GRACE-DS product is propagated through from ocean reanalysis output (Sect. 2.3, Gou et al., 2025). This limits the validity of the comparison in Fig. 2 at the mesoscale. We nevertheless find a common patch of low variances ratios ($R \approx 0.3$) in the

**Table 2.** Global median variance ratios, $\overline{R}$, of the CMIP6-HR models relative to GRACE-DS and BPRs[a]

| Model Name | GRACE-DS, deep | | GRACE-DS, shelf | | BPRs[b] |
| | RMS > 3 cm | RMS ≤ 3 cm | $|\phi| > 60°$ | $|\phi| \leq 60°$ | |
| --- | --- | --- | --- | --- | --- |
| ECMWF | 1.58 (1.40) | 0.71 (0.66) | 1.37 (1.41) | 1.10 (1.03) | 1.02 (0.59) |
| HadGEM | 1.50 (1.47) | 0.81 (0.74) | 1.95 (1.66) | 1.30 (1.34) | 1.01 (0.68) |
| CNRM | 1.15 (1.09) | 0.72 (0.61) | 1.43 (1.36) | 0.91 (1.00) | 0.81 (0.59) |
| CMCC | 1.75 (1.58) | 0.73 (0.57) | 1.38 (1.30) | 1.15 (0.95) | 0.77 (0.47) |
| Earth3P | 1.29 (1.23) | 0.90 (0.73) | 1.44 (1.40) | 1.26 (1.30) | 1.28 (0.74) |

[a] For each model, we list the global median value of $R$, relative to GRACE-DS, in four regions, two regions each in deep water (depths > 1000 m) and shallow water (depths < 1000 m). The partitioning in the deep ocean is by model RMS (> 3 cm or ≤ 3 cm), while shelf regions are separated by latitude $\phi$ (higher or lower than 60°). Values in brackets are the same estimates for the non-seasonal signal content. The last column shows the median variance ratios relative to BPRs.

[b] When compared to BPRs, GRACE-DS has a median variance ratio of 1.30 for the full time-series, and 0.92 for non-seasonal $p_b$ changes.

Northwest Atlantic south of Grand Banks, where the $p_b$ signature of eddies in the GRACE-DS product spread considerably
further into the Atlantic interior (Fig. 1a) than in the CMIP6-HR models. The situation bears close resemblance to Figure 8
in Bernard et al. (2006), which suggests that data-free NEMO simulations on the ORCA025 grid typically underestimate the
magnitude and spatial spread of eddy kinetic energy in the Gulf Stream Extension.

The quantitative results above can be tested for robustness by replacing GRACE-DS with either the CSR or JPL mascons
as reference data (Supplementary Figs. S2–S3, Supplementary Tables S1–S2). These analyses suggest a spatial partitioning of
variance ratios that is very similar to Fig. 2. We again observe a tendency for $R < 1$ in many quiet deep-ocean regions and
Southeast Asian seas, whereas values of $R > 1$ are common in the Southern Ocean, Northwest Pacific, and on continental
shelves in polar latitudes. Among the more prominent deviations from Fig. 2 (GRACE-DS) and Supplementary Fig. S2 (CSR)
are increased variance ratios with JPL throughout most of the Atlantic. These differences hint at uncertainties in the gravity field
recovery in regions of small signal levels. In addition, the patches of red color are more saturated in the evaluation against CSR
mascons, which lack signal content on sub-grid GRACE scales (Fig. 1). Hence, the estimates of $\overline{R}$ from CSR in energetic deep
regions in Supplementary Table S1 cannot be taken at face value. Variance ratios of the coarsened CMIP6-HR fields relative
to the 3° × 3° JPL data offer a cleaner form of comparison, and that analysis does indeed confirm many of the numerical
values and model rankings noted in relation to Table 2 (column 'deep'). On shelf regions, the situation is mostly reversed,
that is, the finer-resolved CSR mascons are better suited as alternative reference data than JPL. Here, Supplementary Table S1
(column 'shelf') again reveals no substantial changes from the corresponding $\overline{R}$ values with GRACE-DS in Table 2. Given
these successful cross-checks, we only employ the downscaled product for the remaining model-data comparisons.

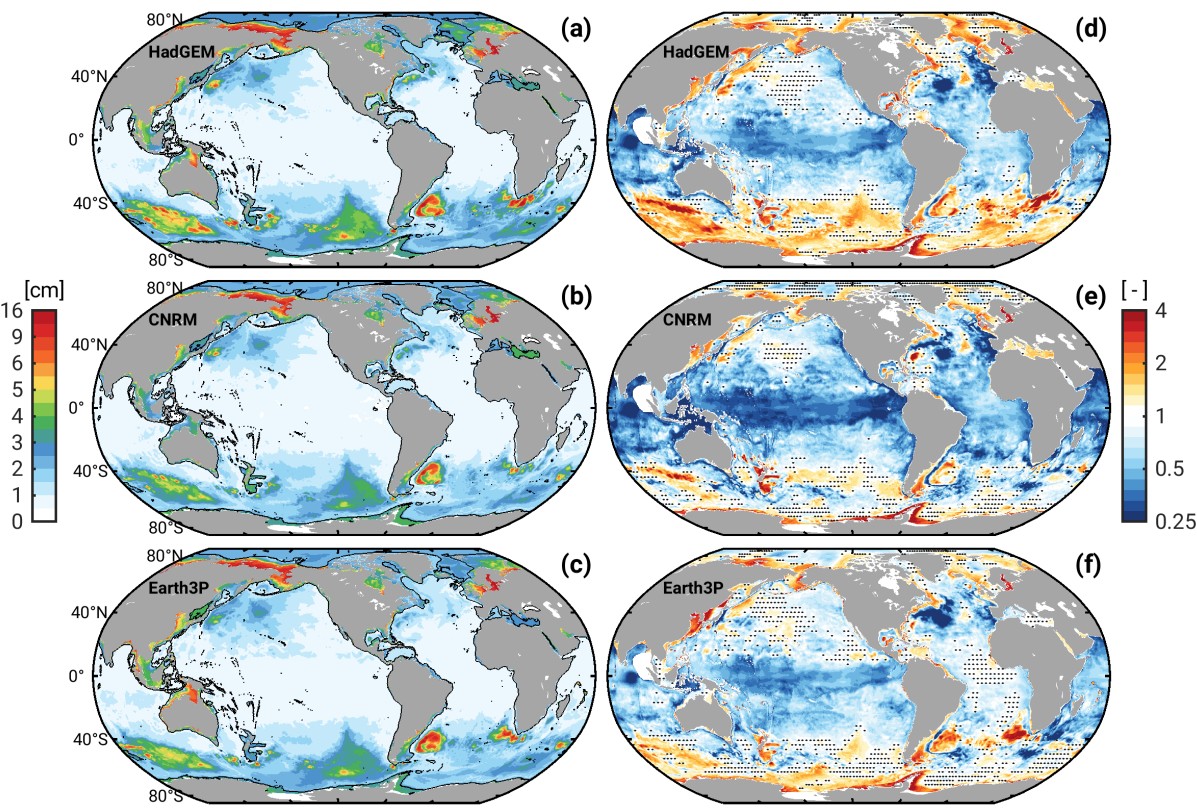

**Figure 3.** As in Fig. 2 but restricted to three models (HadGEM, CNRM, Earth3P) and non-seasonal signals.

### 3.3 Non-seasonal $p_b$ variability

Modeled $p_b$ variability with annual and semiannual oscillations subtracted is briefly assessed for three of the five available models. We select CNRM and Earth3P, which compare favorably to GRACE-DS for energetic deep-ocean regions in Fig. 2,

and HadGEM—a somewhat arbitrary choice but still a counterpoint to the other two models. Global median variance ratios at non-seasonal periods for all five models are compiled in Table 2. As evident from Fig. 3, removal of the seasonal cycle reduces RMS values in deep waters of temperate latitudes to ∼1 cm. At these magnitudes, comparisons with GRACE-DS are likely limited by data noise and subtle systematic errors (cf. Sect. 3.2). These issues inhibit more solid inferences as to why variance ratios in quiet deep-ocean regions decrease relative to the full time series (Table 2, Fig. 2f–j vs. Fig. 3d–f).

Non-seasonal signal levels in many of the more energetic areas in Fig. 3 remain comparable to the full variability, and so do the variance ratios over GRACE-DS. Obvious examples are the Arctic Ocean and Nordic Seas, the Bellingshausen Basin, and several eddy-active regions (e.g., Zapiola Drift, Agulhas leakage, Gulf Stream Extension). By contrast, RMS values for non-seasonal variations are clearly smaller than their broadband counterparts (Fig. 2) in the Gulf of Thailand, Northwest Pacific, Australian-Antarctic Basin, and Weddell-Enderby Abyssal Plain. These changes go along with an improvement of the

variance ratios relative to GRACE-DS toward $R \approx 1$ (even for HadGEM), suggesting that wind-driven geostrophic variability in abyssal-plain regions is indeed too energetic at seasonal periods in CMIP6-HR. Over mid- and low-latitude continental shelves, estimates of $R$ over are largely unaltered as we switch from full to non-seasonal time series, and that behavior is also reflected in the global median statistics (Table 2).

### 3.4  Intermediate discussion

What are potential reasons for the documented differences between modeled and satellite-based $p_b$ variability? As alluded to in Sect. 3.2, the dominance of $R < 1$ in quiescent ocean regions may be attributable to a mixture of data noise, residual systematic effects in the GRACE-DS fields, and mismodeled or entirely absent $p_b$ signals in the simulations. One specific process disregarded in CMIP6-HR are the effects of gravitational attraction and loading (GAL, Vinogradova et al., 2011). GAL induces ocean mass changes in the order of 0.5 cm, particularly at the annual frequency and near continental boundaries

(Ponte et al., 2024). Parts of the higher variance in GRACE-DS in the Northern Indian Ocean and off the Amazon Shelf, which adjoin regions of large terrestrial water storage variations, may be due to GAL. We also note that the low model $p_b$ amplitudes relative to GRACE-DS in the deep-water tropics and some eastern boundary regions (Fig. 2f–j) coincide with areas where $p_b$ variability is influenced by non-linearities and coupling between baroclinic and barotropic modes (Piecuch et al., 2015; Hughes et al., 2018; Zhao et al., 2021). Proper depiction of these processes in the models may pose a challenge and may depend on

spatial resolution or certain parameter settings concerning, e.g., the viscosity of the flow.

Deep energetic regions, on the other hand, show a propensity for too vigorous $p_b$ signals ($R > 1$). Given the importance of wind stress curl and bottom topography in shaping the dynamics of these deep basins (cf. Sect. 3.1), misrepresentation of either of these factors could reduce the fidelity of the modeled barotropic $p_b$ variability. For example, both the horizontal and vertical model resolution may be too coarse to produce the necessary topographic steering and dissipation by form stress

(Weijer and Gille, 2005). Other suspects are imperfect wind stress parametrizations or overly strong surface winds as simulated by each model's atmosphere. Similarly, the enhanced $p_b$ variations on several continental shelves (e.g., in the Arctic) at both seasonal and non-seasonal periods point to issues regarding model topography and dissipation mechanisms. In particular, as the barotropic ocean response to wind stress forcing scales with the inverse of both bed friction and water depth (Csanady, 1982), large $p_b$ variations in shelf regions can occur if the models are too shallow or not frictional enough. These processes may also

contribute to the inflated modeled $p_b$ variability in the shallow parts of the Weddell and Ross Seas (Figs. 2–3). An arguably more critical factor, though, is the blocking of Antarctic ice shelf cavities in CMIP6-HR. This modeling choice constricts the solution to a narrower body of water than in the real ocean and necessarily distorts the dynamics of the region.

### 3.5  Comparison with in situ observations

Comparisons against BPRs in terms of signal levels (RMS), variance ratios, and median variance ratios over all 23 sites are

illustrated in Fig. 4 and Table 2. The analysis complements the previous validation against GRACE-DS in relatively quiet (RMS $\leq 3$ cm) deep-ocean regions, emphasizing in particular the Pacific; cf. the BPR locations in Fig. 1b. For additional context, we also include (but do no extensively discuss) point-wise statistics from GRACE-DS in Fig. 4. A first general observation

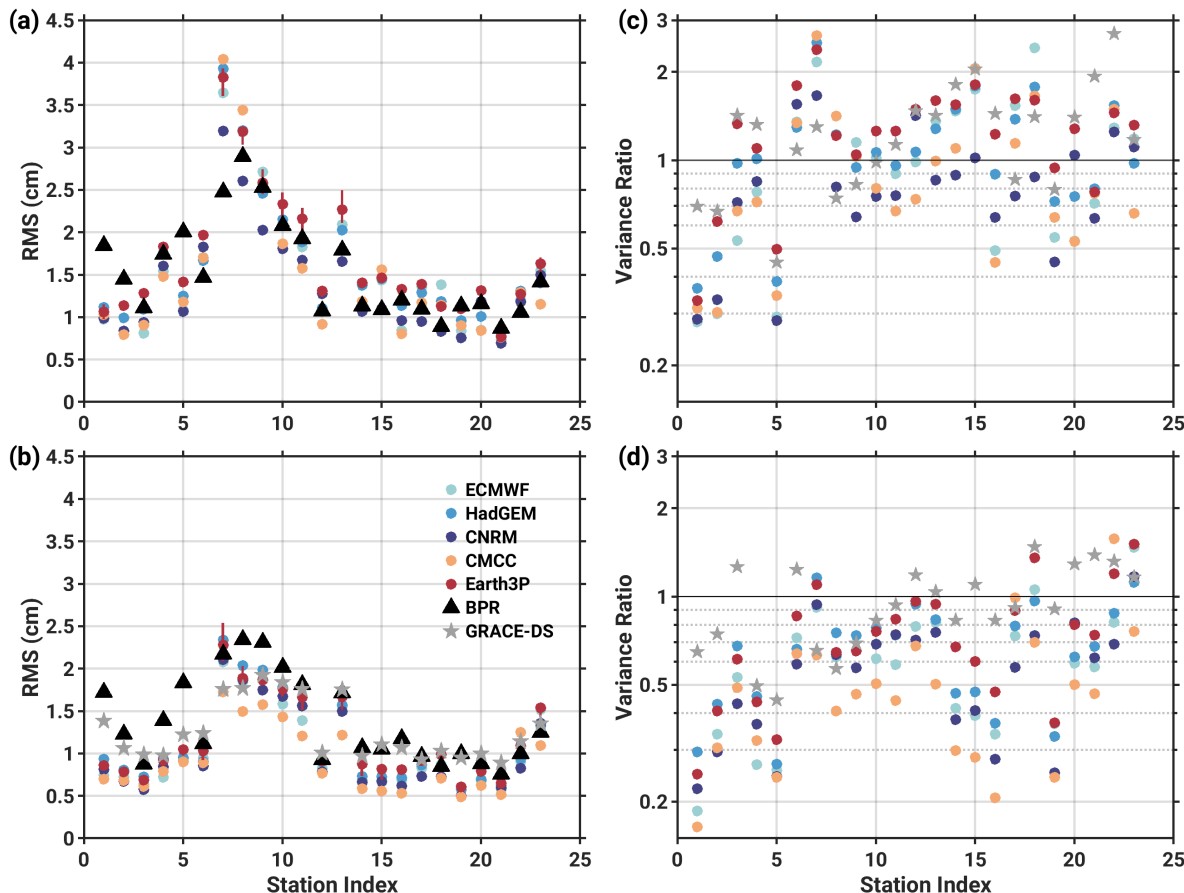

**Figure 4.** Comparison of the CMIP6-HR and GRACE-DS $p_b$ variability against BPR observations at 23 locations (see Fig. 1b) in terms of RMS (left panels) and variance ratios (right panels, $R = 1$ indicated by horizontal gray line). For context, the left panels also include the 10th–90th percentile interval of the RMS from one model (Earth3P, red vertical bars). The top row shows the statistics for the full broadband $p_b$ variations, while the bottom row is for non-seasonal signals.

is that the BPRs have signal levels very similar to those present in the model-based and GRACE-DS $p_b$ estimates at the tested locations. Exceptions to this behavior are rather nuanced and also depend on whether unfiltered or non-seasonal signals (Fig. 4a–b) are analyzed. Specifically, we can discern a tendency for higher in situ than model $p_b$ variance for several sites near the equator (IDs 1–5 and 19), i.e., latitudes where baroclinic and partly non-linear dynamics govern bottom pressure changes (Piecuch, 2015; Hughes et al., 2018). The phenomenon appears to be common to both the full and filtered time series at most of the noted stations.

Somewhat more puzzling, at least at first glance, is that for all sites in the North and Northeast Pacific (IDs 7–16, and partly elsewhere), the model variance ratios hover around 1–1.5 with the seasonal cycle included (Fig. 4c) but drop well below $R = 1$ once it is eliminated (Fig. 4d). This contrast points to systematic differences between BPRs and models in their representation

of seasonal $p_b$ signals. At the points analyzed here, seasonal oscillations evidently make up a larger fraction of the overall $p_b$ variance in the models than they do in the DART data. We suspect that at a resolution of 1/4°, the model bathymetry is too coarse (or too smooth) to capture the damping effects of small-scale topography on the $p_b$ response to seasonal wind stress forcing; cf. Chen et al. (2023) for an in-depth study of the problem in the North Pacific Ocean. Accordingly, the apparent agreement between CMIP6-HR and the BPRs in the broadband case (Fig. 4a) may be partly coincidental; that is, anomalous seasonal variability in the models matches the variance due to non-seasonal local $p_b$ fluctuations in the in situ series. Twin experiments with NEMO in the ORCA025 configuration, employing either a standard 1/4° batyhmetry or smoothed versions of it, could be used to test this hypothesis.

Adopting a more quantitative view, median variance ratios $\overline{R}$ over all BPRs range from 0.77 (CMCC) to 1.28 (Earth3P) and decrease to $\overline{R} = 0.47$–0.74 after removal of the seasonal cycle (Table 2). The comparison of CMIP6-HR with GRACE-DS on a global scale suggests, in fact, very similar ranges and model rankings. This is a reassuring result that alleviates some of our concerns as to the quality of the satellite-based $p_b$ solutions at low signal levels (Sect. 3.3). Consistency in detail is particularly seen along the North and East Pacific margins (IDs 9–20), where both GRACE-DS and the BPRs attribute lowest variance ratios to either CNRM and CMCC and highest variance ratios to Earth3P (Figs. 2–4). Thus, the DART series offer a useful point of comparison, despite their limited spatial coverage and the possible presence of short-scale signals in the records. Joint consideration of the lumped statistics in Table 2 and the GRACE-DS variance ratio maps (Figs. 2–3) nevertheless suggest that Earth3P outperforms the other models in most dynamical regimes. CNRM is least affected by the common issue of excess $p_b$ variance in deep energetic regions, whereas HadGEM comes close to Earth3P in quiescent areas. We proceed with these three models for the remainder of the analysis and note that while other choices would have been possible, the overall $p_b$ characteristics in all five models are still very similar.

### 3.6  Projected changes in $p_b$ variability

We now examine whether the CMIP6-HR models suggest any notable changes in $p_b$ variability under anthropogenic climate change (Fig. 5). To that end, we compute variance ratios, $R$, of the detrended $p_b$ series from the three previously selected models over the time span 2030–2049 (highres-future) with respect to 1980–1999 (hist-1950 simulations). The mid-points of the two windows are therefore 50 years apart, representing the maximum separation possible with the data at hand. We additionally consider 2030–2049 vs. 1980–1999 variance ratios from the control run (Fig. 5, left column), which was integrated for 150 years with fixed forcing (Haarsma et al., 2016). These estimates give an approximate idea of the magnitude of regional $p_b$ changes that emerge internally within the simulations in the absence of time-evolving climate conditions. Evidently, the control run variance ratios from the three models have very little in common and are mostly restricted to the range $R = 0.7$–1.3, with only a few broader statistically significant changes (e.g., in the Bellingshausen Basin for HadGEM). These findings provide useful context for to the discussion of climate-driven changes in the simulated $p_b$ variance (see below). A region that needs to be interpreted with caution, though, is the deep Arctic Ocean, which sees strengthened $p_b$ variability in the control run of all three models. However, that increase is statistically significant only in CNRM.

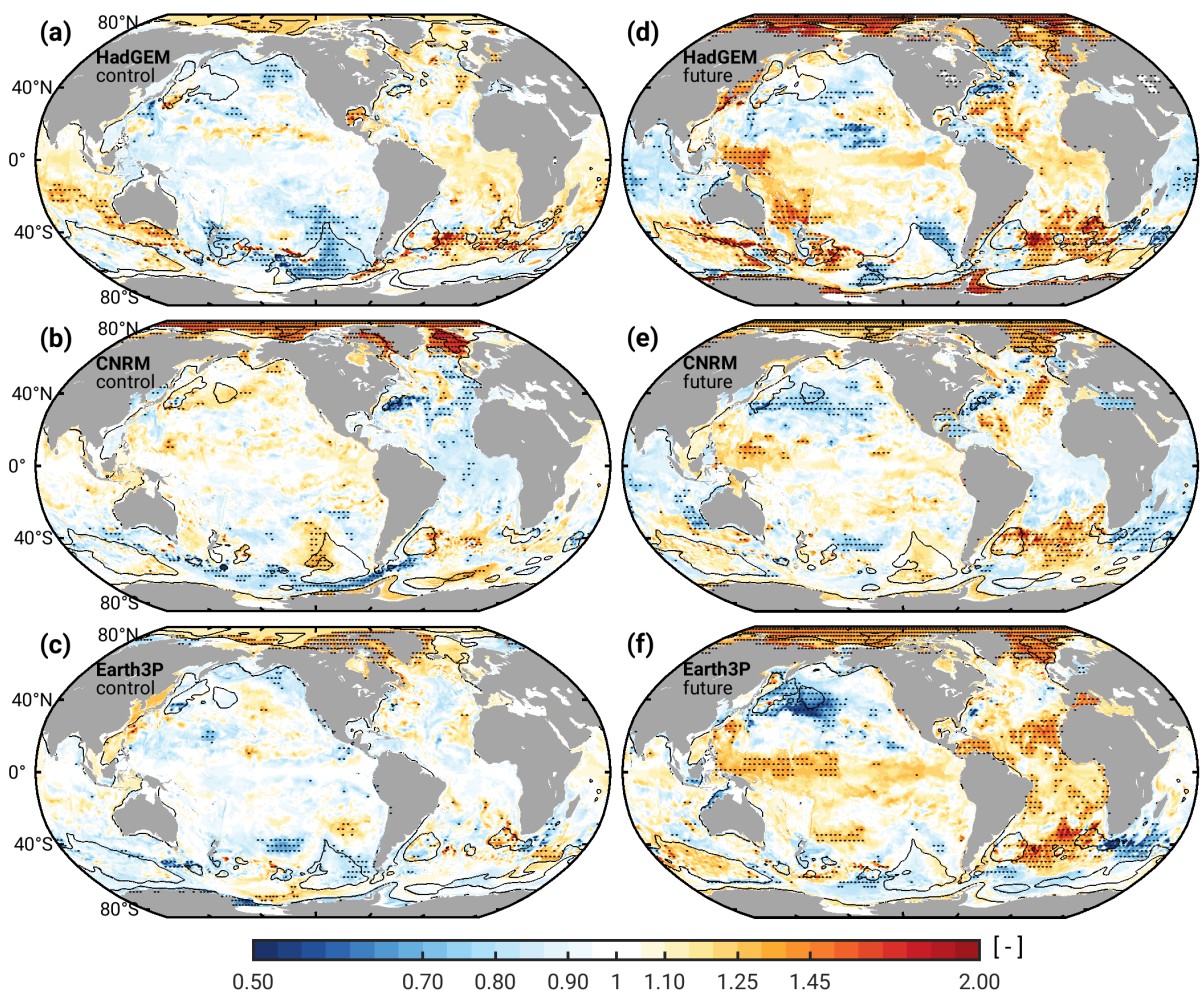

**Figure 5.** Ratio of $p_b$ variance over the time period 2030–2049 relative to 1980–1999 for HadGEM (top row), CNRM (middle row), and Earth3P (bottom row). Left panels (a–c) present the variance ratios from each model's control simulation, whereas the right panels (d–f) show the ratios formed from the scenario and historical simulations. Stipples indicate $3° \times 3°$ cells where the variance ratio is significantly different from zero at 90% confidence for at least a third of the contained $1/4°$ grid points. To determine these confidence levels, we used an $F$-test based on the effective degrees of freedom of the two $p_b$ series compared at each grid point. Black contours in all panels encase areas where the model RMS in Fig. 2, smoothed to 200 km length scales, exceeds 3 cm. Note that the color scale is logarithmic from $-0.3$ to $0.3$, discretized at steps of 0.02.

The maps of $R$ formed using the CMIP6-HR historical and scenario runs (Fig. 5, right column) paint a relatively complex picture of future changes in $p_b$ variance. Amplitude decreases typically alternate with increases on sub-basin scales, involving changes in regions with distinct dynamical regimes. Although the three maps show many differences in detail, we can still discern a few common, statistically significant features. One region of interest is the connected deep basin of the Arctic Ocean

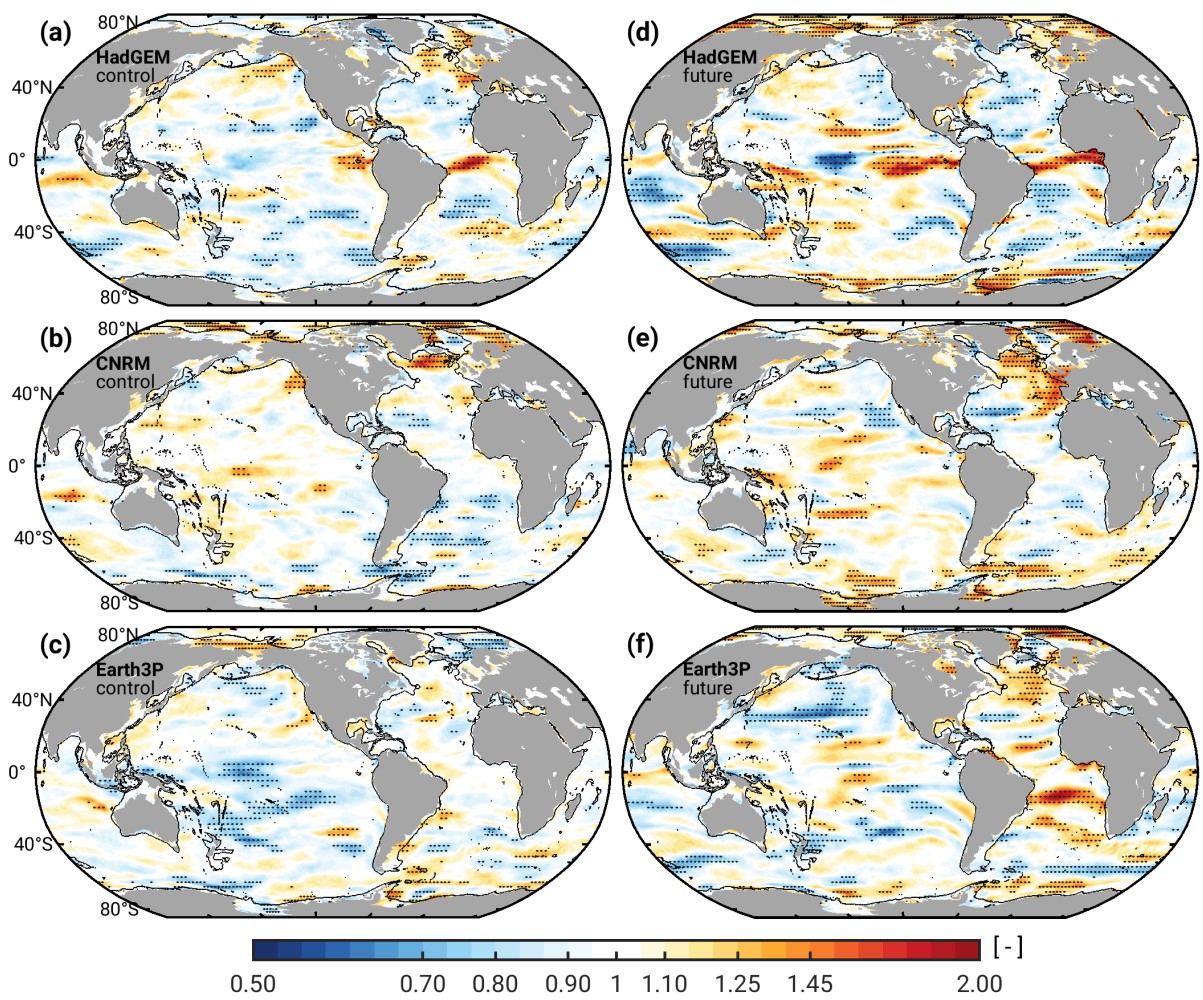

**Figure 6.** As in Fig. 5 but for the variance ratio $R_V$ of the near-surface wind speed (i.e., the magnitude of the horizontal wind vector). Ratios are again formed for the period 2030–2049 relative to 1980–1999. Time-mean wind speeds do not contribute to the picture. To help pinpoint wind speed changes near the continental slope, the 1000-m isobath is included.

and Nordic Seas, with widespread variance increases of 40–60%, partly also extending to adjacent continental shelves. The
increase in CNRM, though, is smaller than in the control run (Figs. 5b and 5e) and therefore questions the results from HadGEM and Earth3P. To examine the matter further—and given the tight connection of $p_b$ fluctuations in the Arctic region to wind forcing (Fukumori et al., 2015)—we compute variance ratios $R_V$ for the near-surface wind speed from the three CMIP6-HR simulations ($R_V = \sigma^2_{2030-2049}/\sigma^2_{1980-1999}$). These estimates are shown for the global ocean in Fig. 6 and in a more focused manner for the Arctic Ocean and its neighboring seas in Supplementary Fig. S4.

A first pass of the $R_V$ maps indicates that while the three models attribute variance gains and reductions to somewhat different geographic locations, the Arctic Ocean and northern North Atlantic are, in general, leading regions for future changes

in near-surface wind speed variability. To link these changes to the modeled increase in Arctic $p_b$ variance, a look at results in Fukumori et al. (2015) is enlightening. Their Figures 9 and 10 reveal the continental slope circling around Northeast Greenland/Canada all the way to the East Siberian Shelf as spatially connected forcing region for the near-uniform Arctic $p_b$

fluctuation. Similarly, the fluctuation can be driven by along-slope winds south of Iceland and the Celtic Sea, but such remote contributions counteract the basin-wide bottom pressure signals excited inside the Arctic domain (Fukumori et al., 2015). This type of cancellation may indeed occur in the CNRM scenario simulation (Fig. 5e), which suggests increases in wind speed variance both in the Celtic Sea and over the continental slope off Northeast Greenland (Fig. 6e, Supplementary Fig. S4e). In contrast, HadGEM, Earth3P, and the CNRM control run see little change in near-surface wind speed variability over remote

forcing regions. Rather, patches of $R_V > 1$ sit over relevant continental slope sections inside the Arctic region, i.e., northeast of Greenland in Earth3P and CNRM-control, and north of the Canadian Arctic Archipelago in HadGEM. Enhanced variability of the associated wind stress could trigger more frequent or more energetic trapped Kelvin waves (Sect. 3.1), thus amplifying the basin-wide Arctic $p_b$ fluctuation in the three models (Fig. 5, panels b, d, f).

Along with some interesting cases of diminishing amplitudes (e.g., Northwest Pacific, Gulf Stream region), we also find

an intensification of $p_b$ variability under SSP5-8.5 in the western tropical Pacific. North of New Guinea, values of $R$ peak at $R \approx 1.3$–$1.5$. The area is characterized by low-magnitude $p_b$ signals (Figs. 1 and 2), which are forced by winds, involve a mix of barotropic and baroclinic processes, and are most active at annual and semiannual frequencies (e.g., Qu et al., 2008; Piecuch et al., 2015; Cheng et al., 2021). Indeed, the patch of intensified $p_b$ amplitudes in the western tropical Pacific vanishes when variance ratios are formed based on non-seasonal model time series (Supplementary Fig. S5). We also note that in Fig. 5, both

HadGEM and Earth3P show increases in $R$ (non-significant) over large parts of the tropical Pacific Ocean. The patterns could hint at broader changes in the wind-driven component of the seasonal cycle around the Inter-Tropical Convergence Zone. A more definite diagnosis of the processes at work is beyond our scope, though.

The third salient feature in Fig. 5, seen across all three models, is a mid-21st century intensification of ∼30–60% in $p_b$ variance in the southern South Atlantic. The patch of red color sits between the Zapiola Gyre and Cape Basin (cf. Fig. 1a) and

is accompanied in eastward direction by a decrease in $p_b$ variability over the Agulhas Return Current. Closer inspection of the corresponding $R_V$ maps (Fig. 6) fails to reveal a common, larger-scale change in wind speed variance that would hint at an origin of the $p_b$ modifications in forced barotropic dynamics. In particular, it is unclear how the east-west dipole in $p_b$ changes centered around the dip of South Africa could arise from variations in surface winds and associated quantities (e.g., wind stress curl).

Instead, we suspect that the $p_b$ structures in Fig. 5 around the junction of the Atlantic and Indian oceans largely bear on the projected changes in the regions's eddy kinetic energy (EKE, Beech et al., 2022; Wang et al., 2024). This is not an unreasonable connection to make, as previous model analyses (Hughes et al., 2018; Zhao et al., 2021) suggest a clear imprint of mesoscale variability on $p_b$ in the areas in question. In terms of processes, global warming is expected to strengthen Southern Hemisphere westerlies and progressively shift them to the south (Deng et al., 2022, see also Supplementary Fig. S6). This change in the

large-scale wind forcing will in turn reduce volume transport and EKE in the Agulhas Current, while also leading to increases in Agulhas leakage and mesoscale activity in the southern South Atlantic (Biastoch et al., 2009; van Sebille et al., 2009; Beech

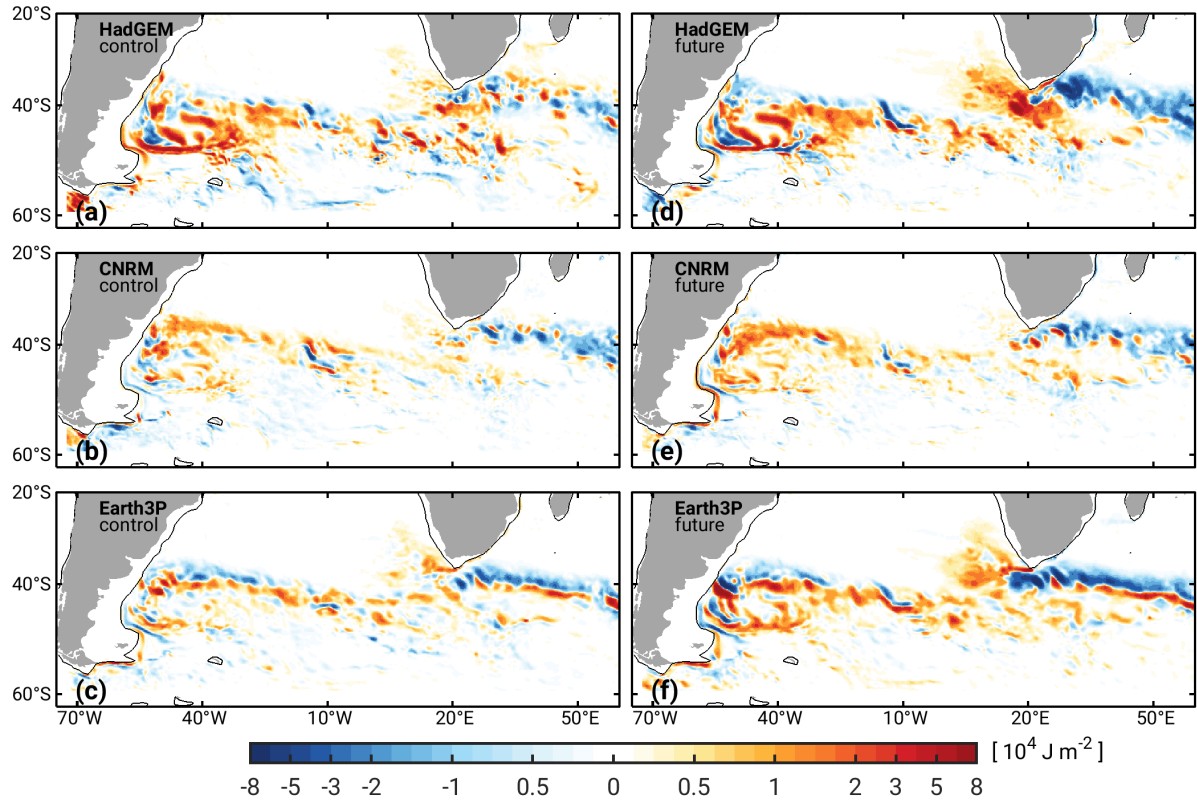

**Figure 7.** Changes in depth-integrated time-mean EKE (J m$^{-2}$), 2030–2049 relative to 1980–1999, around the southern South Atlantic. As in Fig. 5, we present results from HadGEM (top row), CNRM (middle row), and Earth3P (bottom row). Left panels (a–c) show the variance ratios from each model's control simulation, whereas the right panels (d–f) show the ratios formed from the scenario and historical simulations. The black solid line is the 1000-m isobath. Note the non-linear color scale.

et al., 2022). In addition, climate change invigorates eddy activity in the western part of the basin (Beech et al., 2022), and that too could contribute to the enhanced future $p_b$ variance seen in Fig. 5.

We have examined the three CMIP6-HR models for some of the described processes and specifically deduced estimates for the depth-integrated EKE (Ni et al., 2023, Sect. S1 in the Supplementary) in the region. Figure 7 illustrates the resultant changes in EKE by 2030–2049 relative to 1980–1999. As evident from the control run (left column), the full-depth EKE is subjected to considerable changes in distribution and magnitude even under constant climatic conditions. The response under SSP5-8.5 (right column) is nevertheless more pronounced, exhibiting also a good degree of consistency among the three models. Shared characteristics are an increase in eddy activity in and west of the Argentine Basin (with signs of a southward shift in HadGEM

and Earth3P) and reduced full-depth EKE along the path of the Agulhas Return Current. Agulhas leakage clearly intensifies by the middle of the 21st century in HadGEM and Earth3P, but has not yet emerged in CNRM. Overall, the EKE changes produced by CNRM are rather subdued, and so are the changes in the model's $p_b$ variance in the analyzed region (Fig. 5e). In

comparing the variations in oceanic EKE and $p_b$, note that there is no prerequisite for an exact spatial correspondence between the two variables, as the mesoscale imprint on $p_b$ is generally spread out wider than eddies themselves (Zhao et al., 2021). We leave the analysis with these pointers and conclude that the southern South Atlantic and Agulhas Return Current region are an interesting laboratory for linking long-term changes in $p_b$ variability with anthropogenic impacts on the ocean circulation.

## 4    Summary and conclusions

We have run basic and mostly global assessments of bottom pressure signals in CMIP6-HR simulations, motivated in large part by the potential use of these fields in satellite gravimetry simulation studies. The climate models capture all relevant $p_b$ phenomena that can be expected to emerge at the given resolution and under the encoded physics, e.g., eddy structures and wind-driven variability on continental shelves and in the deep ocean. Model accuracy, as evaluated through comparisons to a downscaled GRACE product and BPR measurements, varies with latitude, depth, frequency, and more generally the specific dynamical regime (Figs. 2 and 3, Table 2). In particular, the CMIP6-HR models tested here are susceptible to overestimation of topographically constrained $p_b$ variability, most notably in the Australian-Antarctic and Weddell-Enderby Abyssal plains and at seasonal frequencies. Similar issues are apparent in NEMO-based ocean reanalyses on the ORCA025 grid (cf. Figure 3 in Gou et al., 2025), pointing to structural errors in that particular ocean model setup. As suggested in Sect. 3.4, the excess variance may have diverse causes, including imperfect momentum transfer schemes, coarse vertical spacing of deep layers, and the artificial blocking of ice shelf cavities. In this light, analysis of $p_b$ variability may be useful to guide model improvements or at least complement the standard tests of near-surface variables and meridional overturning (e.g., Zuo et al., 2017; Roberts et al., 2019, 2020).

We have exercised a degree of caution in our comparisons of the CMIP6-HR $p_b$ fields against GRACE-DS and BPRs. Neither reference dataset is free of error, whereas the models only approximate reality and thus disregard several legitimate contributions to the observed $p_b$ changes (e.g., GAL, barometric pressure-driven dynamics, sub-grid scale variability). These issues can compromise the model-data comparison, particularly in very quiet deep-ocean regions (Fig. 3) and at individual BPRs (Fig. 4). The spatial median statistics in Table 2 are nonetheless conclusive and internally consistent enough to discriminate more accurate models from less realistic $p_b$ representations. Specifically, in satellite gravimetry simulations, where one would want to portray oceanic mass changes across all spatial and temporal scales, the HighResMIP contribution of Earth3P appears to be a very sensible choice. One minor caveat is that compared to the models (RMS distribution analyzed over 1980–2014), the BPR and GRACE-DS variance estimates are taken over different time periods, i.e., 2013–2023 and ~2002–2020. However, when narrowing the model time series to 2002–2014 (Supplementary Table S3), our basic conclusions drawn from Table 2 remain unchanged, except for low-latitude shelf regions. The agreement is not unexpected, because the climatic conditions over the CMIP6-HR late historical period evolve less drastically than they do in the scenario extension to 2049, implying temporal changes in $p_b$ variance that are well below those shown in Figs. 5d–f.

Our model-based diagnosis of future climate impacts on the strength of $p_b$ fluctuations has added a new facet to the existing bottom pressure literature. While these impacts are spread over regions characterized by very different dynamics (i.e., Arctic,

tropical Pacific, Agulhas leakage), an origin in altered wind forcing is likely common to all three cases. Dedicated modeling work, such as stand-alone ocean simulations under distinct forcing assumptions, will be required to shed light on the exact processes at work. More generally, the reported strengthening of regional $p_b$ variability awaits confirmation through analysis of output from other climate models. Consideration of a larger ensemble of simulations would also help better distinguish
climate-induced $p_b$ changes from model internal variability (cf. the case of CNRM in Sect. 3.6). At some point it might even be possible to detect the projected $p_b$ changes in actual observations. Here we have made our inferences based on variance metrics in bi-decadal windows 50 years apart, but the noted $p_b$ amplitude increases in the Arctic, tropical Pacific, and parts of the South Atlantic also emerge when using a reference periods shifted forward by 15 years (1995–2014, Supplementary Fig. S7). Thus, given the planned continuity of space-based mass change monitoring (Daras et al., 2024), climate-driven changes in
large-scale, dynamic $p_b$ variability may be soon detectable with satellite gravimetry.

*Data availability.* The datasets used in this study are available from the following links: CMIP6 HighResMIP output (https://esgf-data.dkrz.de/search/cmip6-dkrz/), GRACE-DS (https://doi.org/10.3929/ethz-b-000686843 Gou et al., 2024), CSR monthly mass grids (https://www2.csr.utexas.edu/grace/RL06_mascons.html), JPL mascon data (https://podaac.jpl.nasa.gov/dataset/TELLUS_GRAC-GRFO_MASCON_CRI_GRID_RL06.3_V4), DART[®] historical data (https://www.ndbc.noaa.gov/historical_data.shtml), and ECCOv4r4b
(https://cmr.earthdata.nasa.gov/virtual-directory/collections/C2129193421-POCLOUD). Monthly $p_b$ fields from all five CMIP6-HR models analyzed in this study have been placed at https://doi.org/10.5281/zenodo.14886818 (Liu et al., 2025).

*Author contributions.* **L. L.**: Data curation, Formal analysis, Investigation, Methodology, Software, Visualization, Writing - original draft, Writing - review & editing. **M. S.**: Conceptualization, Funding acquisition, Methodology, Supervision, Validation, Writing - original draft, Writing - review & editing. **L. B.**: Software, Writing - review & editing. **J. F.**: Formal analysis, Software, Writing - review & editing. **J. G.**:
Methodology, Writing - review & editing.

*Competing interests.* The authors declare no conflict of interests.

*Acknowledgements.* This study was supported by the German Research Foundation (DFG, Project nos. 388296632 and 459392861), and represents a contribution to the New Refined Observations of Climate Change from Spaceborne Gravity Missions (NEROGRAV) project. We are grateful to the two reviewers for their helpful comments.

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
