# Peer review of "Assessment of Ocean Bottom Pressure Variations in CMIP6 HighResMIP Simulations"

_EGUsphere, 2025_

## Author Comment (AC1)

**Comments by Referee #1:**

(Referee comments are in black, while our responses are in blue)

A review of "Assessment of Ocean Bottom Pressure Variations in CMIP6 HighResMIP Simulations" by Liu, Schindelegger, Börger, Foth, and Gou

The authors compare ocean bottom pressure (OBP) variability from GRACE/GRACE-FO retrievals, bottom pressure recorder (BPR) observations, and CMIP6 HighResMIP Simulations for past and future periods. They identify regions where observed and simulated OBP time series agree or not, and they also highlight where simulated OBP variability changes substantially from the present to the future, offering interpretations in terms of physical oceanographic processes or observational considerations. The analysis largely focuses on periods from a couple months to a decade, and the authors perform their calculations both with and without the mean seasonal cycle removed.

This is a nice paper that presents a solid incremental advance in the science. As the authors explain, changes in OBP can arise from a variety of oceanographic and geodetic processes, so a study like this will be of interest to a wide community of geoscientists. What's more, the paper is well written, the methods and approaches are generally sound, and most scientific inferences are reasonable and justified. The paper should be published after minor revisions to address a few places where the reasoning could be clarified or the analyses could be expanded to make a stronger study. I thank the authors for making a satisfying study on a topic that's largely been overlooked in papers analyzing past and future climate model simulations.

Good luck,

Chris Piecuch, Woods Hole

**General comments**

Section 2.2 on calculation of OBP anomalies. Why don't the authors use the standard CMIP diagnostic output for OBP (pbo), which is readily available? I'd recommend them to use the proper model diagnostic output because, as the authors explain, right now they're making various assumptions in their calculation of OBP. The errors they're incurring from these assumptions are unclear. First, they're computing density from monthly temperature and salinity using the McDougall and Barker (2011) equation of state. While this sounds reasonable, it risks using an equation of state that's potentially distinct from what the various models use on line. It's also complicated by the nonlinearity of the equation of state (the monthly average of a density time series computed from instantaneous temperatures and salinities is not the same as the density computed from monthly averages of temperature and salinity time series). Second, the authors base their bathymetry on ETOPO1. Again, this sounds reasonable, but model bathymetry can be modified relative to something like ETOPO. Using standard model output alleviates these issues.

Thanks for the commentary. We analyzed two models at the start of the study: ECMWF-IFS 1/4° (see manuscript) and CESM1-3-HH 1/10° (which we dropped along the way). Neither

provides diagnostic 'pbo' output. We therefore set up our own calculation and applied it to other models without checking again for 'pbo'. To follow your suggestion, we would either have to remove ECMWF-IFS from the paper or allow for a mix between model-true and inferred bottom pressures – neither approach is ideal. The good news is that the differences in RMS between 'pbo' and calculated bottom pressures are very small, visually barely distinguishable from each other. To assure readers, we have added the following figure to the Supplementary (Fig. S1):

[Figure]

***Figure S1****: Plots on the left show the RMS (cm) of $p_b$ time series from HadGEM and Earth3P, computed from the respective 'pbo' output over 1980–2014. Plots on the right show the variance ratio relative to the $p_b$ variance from GRACE-DS. Stipples mark 3° x 3° cells where the RMS of GRACE-DS is within the 10th–90th percentile range of the model RMS for at least a third of the contained ¼° grid points.*

In eddy-active regions (e.g., Zapiola Gyre), the 'pbo' RMS is 0.2–0.4 mm smaller than the RMS of the calculated bottom pressures. Outside these regions, differences rarely exceed 0.1 mm. The availability of 'pbo' fields and the small differences seen in Fig. S1 are now mentioned at the end of Section 2.1 in the main text.

Section 2.3 on downscaling GRACE. I'd like the authors to discuss why they downscale GRACE/GRACE-FO data rather than coarsen CMIP6 model output. The downscaling is based on machine learning algorithms that incorporate eddy-permitting ocean circulation models. It's unclear what the associated uncertainties are. Are the downscaled datasets also based on the NEMO modeling framework? If so, then we have the potential of correlated errors between the downscaled GRACE/GRACE-FO products and the CMIP6 models. I'm not recommending the authors fundamentally change their approach. But I would like them to (1.) justify their decision to downscale rather than coarsen and (2.) discuss the associated uncertainties, biases, and other potential implications.

We have addressed this issue by repeating the model-data comparison ('full' time series) with two genuine GRACE datasets: (i) the CSR mascons that underlie the downscaled GRACE

product, GRACE-DS, and (ii) the coarse JPL mascons. For (ii), we averaged the CMIP6-HR fields to the exact same grid used by JPL (3° equal-area caps). Note that such coarsening is not feasible when comparing to the CSR mascons, because CSR employs a stochastic process model in the gravity field estimation that is difficult to mimic by simple spatial averages. The results of the new analyses are presented in Supplementary Figs. S2–S3 (variance ratio maps) and Supplementary Table S1–S2 and are briefly discussed at the end of Section 3.2 in the main text ('Comparison with GRACE – broadband variability'). We include the middle part of Fig. S3 in this reply for illustration purposes:

[Figure]

***Figure S3***: *Variance ratios of the full CMIP6-HR $p_b$ time series from HadGEM, CNRM, and CMCC, with JPL mascons chosen as reference data and the model $p_b$ fields averaged to the coarse JPL grids. Stipples mark 3° x 3° cells where the RMS of the observations is within the $10^{th}$–$90^{th}$ percentile range of the model RMS.*

In short, the validation of the coarsened model output against JPL confirms the previously noted overestimation (underestimation) of modeled OBP variability in energetic (quiet) deep-ocean regions. Global median variance ratios, $\bar{R}$, for the energetic case are essentially identical to estimates of $\bar{R}$ with GRACE-DS as reference. For the quiet deep-ocean case, JPL suggest $\bar{R}$ values that are somewhat (~0.1) higher than with GRACE-DS. We attribute this difference to uncertainties in the various gravity field solutions (CSR, JPL) in the Atlantic; it is not something the downscaling introduces. On continental shelves, the original CSR mascons are an alternative benchmark, and here again, we observe only small changes (0.01 to max. 0.15) in the variance ratio statistics compared to the evaluation against GRACE-DS.

To prepare for these analyses in the text and motivate them in light of the possible limitations of GRACE-DS, we have expanded Section 2.3 ('GRACE data'). We have also modified Fig. 1 to show, along with the GRACE-DS RMS, the RMS of the CSR parent solution. This visual comparison allows one to (i) grasp the OBP variability added by the downscaling to the GRACE data, and (ii) appreciate the similarity between the two datasets at typical GRACE wavelengths. The justification for using GRACE-DS as main validation data simply reads "… we focus on GRACE-DS, as its signal content and characteristics are most commensurate with the CMIP6-HR models across scales and from deep to shallow regions."

Equation 2 (the variance ratio). The authors define this as the modeled variance divided by the observed variance. When they show results on a linear vertical scale, this definition will tend to visually overemphasize values >1 and de-emphasize values <1 (i.e., the former will span a greater color range than the latter). Therefore, I'd suggest, whenever they're showing R values, the authors use more of logarithmic color scale or vertical axis. That way, values >1 and <1 would visually communicate comparable emphasis.
Very good idea; we have implemented this in all figures showing variance ratios.

In all the figures, the authors compare root mean square (RMS) variability from the models and the observations. Typically, modeled RMS values are computed for a fixed time period (e.g., 1980-2014). However, Figure 5 shows the very interesting and important result that, even under a control simulation with presumably stationary statistics, there can be large apparent changes in RMS amplitudes (see left column). Because of this stochastic variability, it's unclear whether any of the R values in the preceding figures are significant or not. Therefore, I'd like the authors to perform a more comprehensive error analysis. Rather than computing modeled RMS values over a single period (e.g., 1980-2014), I'd suggest the authors instead compute RMS values for overlapping but separate periods to approximate a distribution of RMS values that would better quantify uncertainty and permit them to test whether simulated OBP variability really is distinct from what we're seeing in the observations.
Thanks a lot for this helpful suggestion. We have added a paragraph to Section 2.2 ('Calculation of ocean bottom pressure anomalies') that introduces the non-stationary character of the model RMS to the reader, along with our approach to reflect this in the presentation. We closely followed your idea of creating an RMS distribution from rolling (in our case 15-year) windows within the longer time period, 1980–2014. In figures that show variance ratios model-versus-GRACE, we stipple grid points where the GRACE RMS is inside the 10[th]–90[th] percentile range of the model RMS. See Figs. S1 and S3 above for examples. In addition, in plots and for all quantifications, we now use the median of the distribution to infer the RMS of a particular model at each grid point (previously: RMS over the full 1980–2014 span). This led to minor changes of the global median variance ratio estimates.

Note that these basic error considerations based on RMS distribution only pertain to the assessment OBP output from historical simulations (Sections 3.2–3.5 in the revised manuscript). For the analysis of projected OBP changes in the models, we had already implemented proper statistical tests (see Fig. 5).

Starting on line 228, the authors note that models show stronger OBP variability on shelves compared to observations. They may mention that this could arise partly if the models aren't frictional enough or are too shallow (e.g., you expect barotropic ocean response to scale with the inverse of both friction coefficient and ocean depth).

Thank you for the insights; we have added this argument to the new Section 3.4 'Intermediate discussion', which we have created to accommodate suggestions by Referee #2 concerning the interpretive framing.

In section 3.5, the authors argue that increased future OBP variability could be related to changes in zonally averaged wind speeds (Figure 6). To me, this is an apples to oranges comparison. Assuming a linear barotropic adjustment such that OBP responds to winds on the same timescale, the more relevant comparison to make here would be to compare RMS (not mean) zonal wind speeds between the two periods. That is, the authors should quantify if the winds will grow more variable in the future (not just if they grow stronger overall).

As for the increase in OBP variance in eddy-rich regions of the South Atlantic, it is meaningful to analyze changes in the overall wind speeds, as their intensification and poleward shift appear to be the root cause for increased Agulhas leakage (e.g., Biastoch et al. 2009, Beech et al. 2022 and references therein). In the Arctic (our other regions where make a serious connection to surface winds), one should look at the wind speed RMS, that is correct. We have done the necessary analysis and created a preliminary plot of changes in wind speed variance for inclusion in the Supplementary. In all coupled model simulations that show a significant increase in OBP variance across the Arctic Ocean and Nordic Seas, we find enhanced future wind speed RMS in some of the forcing regions for Arctic OBP variability (see Fukumori et al. 2015).

**Line edits:**

Line 34: Suggest to delete "mainly because they are arbitrary in time"
Phrase deleted.

Line 50: Suggest to delete "by us"
Done.

Line 51: Suggest to change "Given the monthly sampling of the data and the fact that model drift precludes the study of trends" to "Given the monthly sampling of the data and the fact that models drift, we are precluded from studying trends"
We now write "Given the monthly sampling of the data and possibility of spurious model drifts, we are precluded from studying trends and exclusively analyze pb signals at intraseasonal to interannual periods."

Line 68: Suggest to change "Ad hoc short names" to "Abbreviations"
Done.

Equation 1: Change "\int_{0}^{\eta} \rho_0 g dz" to "\int_{0}^{\eta} \rho g dz" and change the second equal sign to an approximation symbol (since the authors make the very reasonable approximation that density is constant over the vertical distance between 0 and sea level)
Done, thank you.

Line 97: Change "sketchy" to "uncertain"
Done.

Lines 173: Specify *planetary* potential vorticity
Done.

Line 184: Suggest to change "baroclinic instability" simply to "instability" to be more general
Done, thanks.

Line 208: Suggest to change "are therefore likely" to "may be"
Done.

Line 262: The concept of geostrophic modes (resonances) is fairly specific (and esoteric; see Greenspan 1968). Suggest to change "geostrophic modes" to simply "variability".
Done, thanks.

Line 274: Suggest to change "from this behavior" to "to this behavior"
Done.

Line 275: pb should be italicized
Done.

---

## Author Comment (AC2)

**Comments by Referee #2:**

(Referee comments are in black, while our responses are in blue)

This paper evaluates ocean bottom pressure (OBP) variability in high-resolution climate model simulations submitted to CMIP6 under the HighResMIP protocol. The authors compare model-derived OBP variance fields at 1/4° resolution with observation-based estimates from downscaled GRACE satellite data and in situ bottom pressure recorders (BPRs). Their results suggest the models overestimate variance in some regions (notably on continental shelves and the Southern Ocean abyssal plains) while underestimating it in more quiescent deep-ocean areas relative to observations. Future scenario analysis indicates a projected increase in OBP variance in high-latitude and eddy-active regions, which the authors link to enhanced wind forcing and intensified Agulhas leakage, suggesting interesting implications for satellite gravimetry and climate change detection.

With new gravity missions planned by ESA and NASA, the paper is timely and addresses a gap in our understanding of how high-resolution climate models represent ocean bottom pressure variability, and will be of interest to the oceanographic and geodetic communities. Overall, the manuscript is well written, the analysis is well-grounded and the figures are of good quality.

A significant concern, however, arises from the choice of reference data used to assess model performance. The GRACE-DS product, while innovative, is downscaled using ocean reanalysis outputs from GLORYS and ORAS5, both of which are based on the NEMO ocean model. Since NEMO also forms the ocean component of all five CMIP6-HR models assessed here, the evaluation may suffer from circularity: structural biases present in NEMO-based models could propagate into both the GRACE-DS product and the CMIP6 simulations. This undermines the independence of the benchmark and makes it difficult to unambiguously attribute over- or underestimation of OBP variance to model error rather than artefacts of the downscaling process. Moreover, the over or underestimations may be greater than given by this analysis. A deeper discussion of this limitation, or a sensitivity test using alternative GRACE products and/or reanalyses, would help clarify the robustness of the findings. This may be beyond the scope of the present work, but potential limitations should be more fully acknowledged. And, in light of these issues, it would be more appropriate to describe the results in the more neutral terms of relative amplitudes rather than the loaded terms 'overestimation' and 'underestimation'. This shift in language would reduce the implication that one dataset is definitively correct and better reflect the comparative nature of the analysis.

We understand the skepticism and agree that potential limitations of GRACE-DS and resultant implications for the analysis were not properly addressed in the original manuscript. The concern largely overlaps with the second point raised by Reviewer #1, so we write a very similar response: As a sensitivity test, we have repeated the model-data comparison ('full' time series) with two genuine GRACE datasets: (i) the CSR mascons that underlie the downscaled GRACE product, and (ii) the coarse JPL mascons. For (ii), we averaged the CMIP6-HR fields to the 3° equal-area caps used by JPL, which allows for a cleaner comparison at typical GRACE wavelengths. The results of these new analyses are presented in Supplementary Figs. S2–S3 (variance ratio maps) and Supplementary Table S1–S2 and are briefly discussed at the end of Section 3.2 in the main text ('Comparison with GRACE – broadband variability').

In short, the validation of the coarsened model output against JPL confirms the previously noted characteristics of modeled OBP variability in energetic and quiet deep-ocean regions. Global median variance ratios, $\bar{R}$, for the energetic case are essentially identical to estimates of $\bar{R}$ with GRACE-DS as reference. For the quiet deep-ocean case, JPL suggest $\bar{R}$ values that are somewhat (~0.1) higher than with GRACE-DS. We attribute this increase mainly to differences in the various gravity field solutions (CSR, JPL) in the Atlantic; it is not something the downscaling introduces. On continental shelves, the original CSR mascons are an alternative benchmark, and here again, we observe only small changes (0.01 to max. 0.15) in the variance ratio statistics compared to the evaluation against GRACE-DS. Overall, these cross-checks support our initial conclusions based on GRACE-DS and allow us to attribute high and low relative amplitudes in CMIP6-HR to model errors, and not to artefacts from the downscaling.

Related points & modifications:
- Section 2.3 ('GRACE data') acknowledges possible limitations of GRACE-DS and prepares for the use of the CSR and JPL mascons in the supplementary analyses.
- Figure 1 has been updated to show the bottom pressure RMS from both GRACE-DS and the CSR parent solution.
- Given the successful checks for robustness, we still use the terms 'overestimation' and 'underestimation' in our presentation. However, we agree that these terms are somewhat loaded and have adopted alternative formulations in about half of the cases.

A broader concern is the paper's tendency to offer speculative explanations for inter-model differences and model–observation mismatches without direct supporting evidence. While many of the proposed mechanisms—such as topographic smoothing, wind stress misrepresentation, blocked ice shelf cavities, or changes in eddy activity—are plausible, they are presented more as assertions than as tested hypotheses. For example, differences in bottom pressure variance are attributed to bathymetric constraints or wind forcing without showing comparative diagnostics of wind fields, bathymetric detail, or eddy characteristics across models. One exception is for the Arctic and South Atlantic where there is a rather superficial attempt to relate long-term changes of OBP variances to the wind field, yet this is not convincing or properly developed. Similarly, interpretations of future OBP variance increases invoke processes like Ekman pumping, Rossby waves, or stratification changes, but these remain unexamined. This interpretive style, while common in model evaluation studies, risks overreaching and may give a false sense of causal understanding.

(comment continued)

I suggest removing the speculative content currently embedded in the results section, collating and synthesising it in a separate discussion section. This would clarify the distinction between empirical findings and interpretive hypotheses and improve the scientific rigour of the paper. While this restructuring might leave the results section relatively brief, it creates an opportunity to deepen the quantitative analysis — for example, by more systematically evaluating inter-model spread, introducing uncertainty estimates for variance ratios, or

providing more regional or temporal breakdowns of the comparisons. A clearer separation between results and interpretation would also help readers better assess the robustness of the conclusions.

Thanks for the critical comment. We agree that our approach of presenting results – starting each paragraph with quantifications and closing with interpretation – was not ideal. We have dealt with this issue as follows:

- We have removed several arguments that were stretching it too far. The invoked processes were coastally trapped waves (Section 3.3 'Non-seasonal pb variability'), suppression of mesoscale signals on the continental slope (previously Section 3.4 'Comparison with in situ observations'), and Ekman pumping, baroclinic Rossby waves, and changes in stratification (previously Section 3.5 'Projected changes in pb variability').
- Other thoughts on possible model limitations that lead to differences with the satellite data (e.g., model resolution, shallow bathymetry, momentum transfer and dissipation schemes, blocking of ice shelf cavities) are now presented collectively in an intermediate discussion (new Section 3.4). After that, we move on to the comparison with BPRs, where one more interpretation (on topographic smoothing) is offered. This is the only manuscript structure that works considering the scope of the paper and the need for a natural flow of the presentation.
- In formulating our interpretations, we now mostly use the subjunctive. This reduces the risk of giving the reader a false sense of causal understanding.

We also would like to note that even if the new intermediate discussion is disregarded, the quantitative results section in the manuscript is not short. Region-specific information is already contained in the figures, while uncertainty information for model-data differences and validation against original GRACE data have been added during this revision. At the time of this writing, we are working on an improved version of Fig. 6 to bolster our arguments about the connection of changes in OBP variance to changes in surface wind speed. Apart from these new elements, we do not intend to add more results or drop individual sections to deep certain aspects of the quantitative analysis.

Given the novelty and relevance of the topic, and the sound core methodology, I believe the paper has the potential to make a valuable contribution. However, the concerns outlined above—particularly regarding the choice of reference dataset and the interpretive framing—should be addressed through major revision.

**Minor comments**

Line 56: "valorizations" -> "evaluations" or similar.
Done.

Line 96: "sketchy" -> "uncertain".
Done.

Line 142: reflecting *net* atmospheric pressure variations over the ocean
Done, thanks.

Line 216: "considerable" -> "somewhat"
Done.

Line 214: "This type of…" - this is an important caveat that needs stating upfront.
We have moved the caveat regarding mesoscale OBP signals in the GRACE-DS product to the data description section (Section 2.3).

Line 250: This is a weak justification.
There is no compelling reason to favor any particular model from the available pool (ECMWF, HadGEM, CMCC). We have changed our justification to "… a somewhat arbitrary choice but still a counterpoint to the other two models."

Line 258: Must be the case.
Agreed, but we have removed/reformulated this passage.

Line 273: Why not discuss in more detail?
GRACE-DS has been previously validated against BPRs in Gou et al. (2025). Moreover, because the manuscript is about OBP signal content in the climate models, such discussion would rather be a digression.

Line 274: "signal levels" -> "amplitudes".
We see nothing wrong with the word "signal levels" here and in other places of Section 3.

Line 275: pb italicised.
Done.

Figure 5: This caption is rather confusing.
We modified some of the wording to improve clarity.

Line 347: Why should decelerated winds lead to a reduction in Agulhas transport and eddy activity?
That was a fallacious argument, indeed. The cited papers (e.g., van Sebille et al. 2009) suggest that the key element is a repositioning of the latitude of zero wind stress curl due to the poleward shift of westerly winds over the Southern Ocean. We have reformulated the passage to something less specific: "… global warming is expected to strengthen Southern Hemisphere westerlies and progressively shift them to the south (Deng et al. 2022, see also Fig. 6). This change in the large-scale wind forcing will in turn reduce volume transport and eddy kinetic energy in the Agulhas Current, …"